# Sugarcane Streak Mosaic Virus P1 Attenuates Plant Antiviral Immunity and Enhances Potato Virus X Infection in *Nicotiana benthamiana*

**DOI:** 10.3390/cells11182870

**Published:** 2022-09-14

**Authors:** Kun Zhang, Xiaowei Xu, Xiao Guo, Shiwen Ding, Tianxiao Gu, Lang Qin, Zhen He

**Affiliations:** 1Department of Plant Protection, College of Horticulture and Plant Protection, Yangzhou University, Yangzhou 225009, China; 2Jiangsu Key Laboratory for Microbes and Functional Genomics, Jiangsu Engineering and Technology Research Center for Microbiology, College of Life Sciences, Nanjing Normal University, Nanjing 210023, China; 3Joint International Research Laboratory of Agriculture and Agri-Product Safety of Ministry of Education of China, Yangzhou University, Yangzhou 225009, China

**Keywords:** expression, four-dimensional label-free quantitation method (4D-LFQ), interactions, pathogenesis-enhancing, photosynthesis, P1 protein, viral suppressor of RNA silencing (VSR)

## Abstract

The sugarcane streak mosaic virus (SCSMV) is the most important disease in sugarcane produced in southern China. The SCSMV encoded protein 1 (P1^SCSMV^) is important in disease development, but little is known about its detailed functions in plant–virus interactions. Here, the differential accumulated proteins (DAPs) were identified in the heterologous expression of P1^SCSMV^ via a potato virus X (PVX)-based expression system, using a newly developed four-dimensional proteomics approach. The data were evaluated for credibility and reliability using qRT-RCR and Western blot analyses. The physiological response caused by host factors that directly interacted with the PVX-encoded proteins was more pronounced for enhancing the PVX accumulation and pathogenesis in *Nicotiana benthamiana*. P1^SCSMV^ reduced photosynthesis by damaging the photosystem II (PSII). Overall, P1^SCSMV^ promotes changes in the physiological status of its host by up- or downregulating the expression of host factors that directly interact with the viral proteins. This creates optimal conditions for PVX replication and movement, thereby enhancing its accumulation levels and pathogenesis. Our investigation is the first to supply detailed evidence of the pathogenesis-enhancing role of P1^SCSMV^, which provides a deeper understanding of the mechanisms behind virus–host interactions.

## 1. Introduction

Sugarcane (*Saccharum officinarum* L.) is an important biomaterial and is widely cultivated in tropical and subtropical regions. In Brazil and India, sugarcane is typically used for sugar extraction and ethanol productions [1]. Sugarcane reproduces asexually, so it is easily infected by various plant viruses that cause diseases, including sugarcane yellow leaf disease [2], sugarcane mosaic disease (SMD) [3], and sugarcane bacilliform virus disease [4]. In India, approximately 50 diseases have been reported in sugarcane [5]. SMD caused by sugarcane mosaic virus (SCMV), sugarcane streak mosaic virus (SCSMV), and sorghum mosaic virus infections dramatically decreased sugarcane production and led to varietal degeneration in the field [1]. SCSMV belongs to the genus *Poacevirus* of the family Potyviridae and has become widespread in the primary sugarcane-cultivating areas of China in recent years [6]. SCSMV is a positive single-stranded RNA virus of approximately 9.8 kb that encodes one large polyprotein. Its mature form contains 10 proteins after separation by hydrolytic cleavage [7]. SCSMV can be mechanically transmitted through infected cuttings during asexual reproduction in the field [8]. SCSMV-infected sugarcane shows various symptoms, such as uneven leaf size, fading leaf colour, and the formation of irregular mosaics, streaks, or mottles [6]. Hence, virus infection severely alters the developmental and physiological state of the host plant and the signal transduction processes (proteins) involved in SCSMV–sugarcane interactions.

Previous studies have demonstrated that the P1 protein of SCSMV (P1^SCSMV^) has a strong RNA silencing suppressor (RSS) activity [9] and several conserved motifs that are positively associated with its RSS activity and protein stability [10]. P1 encoded by Triticum mosaic virus (TriMV), which also belongs to the genus *Poacevirus*, can suppress the host antiviral RNA silencing pathway through double-stranded RNA (dsRNA) binding activity [11]. As a virus-encoded RNA silencing suppressor (VSR), the P1 of *Poacevirus* could act as a disease-enhancer in the *cis*-heterogeneous expression of the potato virus X (PVX) in *Nicotiana benthamiana* [4]. VSRs often directly or indirectly inhibit or modulate plant innate immunity to achieve the multifunctional roles in virus–plant interactions [12,13,14]. However, how P1^SCSMV^ alters the physiological states and signal transduction processes (proteins) of the host plant is still obscure.

Recently, significant advances in large-scale, high-quantification accuracy and dynamic range technologies have been achieved. They include label-free peptide quantitation with enhanced ion mobility, also called four-dimensional (4D) proteomics [15,16]. Owing to the low sample requirement, high accuracy, and wide proteome coverage, the 4D-proteomic approach has become one of the primary tools in functional genomics studies in different research fields [17,18,19,20]. In agricultural science, label-free peptide quantitation proteomics has been widely used to characterise plant responses to abiotic and biotic stresses, including stresses caused by plant virus infections [21,22]. Due to a lack of the infectious clone of SCSMV, the PVX-mediated overexpression system is well-characterised, with high protein expression levels, systemic infection, and persistence in *N. benthamiana* [23,24]. Thus, the PVX system was utilised to analyse the response of viral-mediated *cis*-overexpression of P1^SCSMV^ compared with that of green fluorescence protein (GFP) in *N. benthamiana*. This experiment was tested using the 4D-proteomic approach.

In the present study, we functionally annotated differentially accumulated proteins (DAPs) that were identified in leaves where P1^SCSMV^ was expressed. The credibility and reliability of the proteomic data were evaluated using qRT-RCR and Western blot analyses. The physiological response caused by host factors that directly interacted with the PVX encoded proteins was more pronounced for enhancing the PVX accumulation and pathogenesis in *N. benthamiana*. Moreover, P1^SCSMV^ attenuates host photosynthesis by damaging PSII. Taken together, P1^SCSMV^ plays a significant role in altering the physiological status of its host by up- or downregulating the expression levels of the host factors that directly interact with the viral proteins. This creates an optimal condition for PVX replication and movement and enhances the accumulation levels and pathogenesis of PVX in *N. benthamiana*. To our knowledge, our investigation provided the first detailed evidence of the pathogenesis-enhancing role of P1^SCSMV^, which will help to form a deeper understanding of the mechanisms underlying virus–host interactions.

## 2. Materials and Methods

### 2.1. Viral-Derived Vector Construction

The P1^SCSMV^ ORF sequence was amplified by the primers pTSK/3′FlagClaI/F (5′-CCATCGATATGGACTACAAAGACGATGACGACAAAGA-3′) and SCSMV-P1/5′SalI/R (5′-GCGTCGACTCAATAAAATACTAAATCTTC-3′). After recycling the target fragment, Cla I and Sal I were used for double digestion. The digested product was ligated to the Cla I-/Sal I-digested viral vector pND108 [25] and thus generated the plasmid PVX-P1^SCSMV^. The original pGreen208 [25] vector was named PVX-GFP.

### 2.2. Plant Growth Conditions and Virus Inoculation

A climate-controlled chamber with 16 h light (~80 mmol/m^2^) and 8 h dark photoperiod at 24 °C was used for the growth of *N. benthamiana* plants. The agrobacteria containing the cDNA clones of PVX-GFP and PVX-P1^SCSMV^ were cultured on Luria Bertani (LB) plates with corresponding antibiotics for 2 days, then transferred to LB broth. After overnight shaking cultivation, the agrobacteria were harvested and suspended in agrobacterium suspension buffer. The final infiltration concentration of PVX-GFP and PVX-P1^SCSMV^ was 0.4 (OD_600_).

### 2.3. Antiserum Preparation

Prokaryotic expression and purification of the recombinant His-CP^PVX^ proteins was carried out using *E. coli* strain BL21, and the obtained proteins were used to immunise New Zealand white rabbits for generation of the polyclonal CP^PVX^-specific antibodies. After five immunisations in 40 days, the antiserum was obtained and the titer was determined. By further precipitation and dialysis, the purified CP^PVX^-specific antibodies were obtained.

### 2.4. Measurement of the Photosynthesis

The influence of SCSMV infection on sugarcane photosynthesis was determined by the GFS-3000 and Dual-PAM-100 measuring devices (Heinz Walz GmbH, Effeltrich, Germany). In the test, the temperature, CO_2_ concentration, relative humidity, and red actinic light were set at 25 °C, 400 ppm, 65%, and 267 micro-mol·m^−2^·s^−1^ in the test, respectively. The dark fluorescence yield (F_0_) and maximum fluorescence (Fm) were certified by darkness treatment of the samples for 30 min. The steady-state fluorescence yield (F) and maximum fluorescence yield (Fm′) were calculated during the light stage. The electron transport rate (ETR) of PSII was determined as described previously [26,27]. The net photosynthesis rate (Pn) and intracellular CO_2_ mole fraction (Ci) were calculated by GFS-3000 and its software, as previously described [27].

### 2.5. Agroinfiltration and GFP Imaging to Investigate the RNA Silencing Suppressor Activity

The plasmids PVX-GFP and PVX-P1^SCSMV^ were transformed to *Agrobacterium tumefaciens* GV3101 by a freeze–thaw method, as described previously [28]. The LB medium (100 mg/L Kan and 25 mg/L Rif) was used for A. tumefaciens culturing. After culturing for 10–16 h at 28 °C by shaking at 220 rpm, the cells were collected by centrifugation at 3000× *g* for 10 min, followed by suspension with infiltration buffer, and incubation at 28 °C more than 2 h prior to infiltration, as previously described [29].

To evaluate the RNA silencing suppression activity of P1^SCSMV^ protein, the *A. tumefaciens*-containing *35S*-driven ssGFP cassettes combined with equal amounts of bacterial suspensions (harbouring plasmids of PVX-GFP and PVX-P1^SCSMV^), with optical density (OD_600_ = 0.5), were inoculated into the abaxial leaves of 6-week-old *N. benthamiana* with a needle-free syringe. After 5 dpi, a long-wave ultraviolet lamp (B-100AP/R, UVP) was used to observe the intensity of the green fluorescence. The plants were photographed by a digital camera (EOS 80D, Canon) with a yellow filter (Gelatin filter No. 15, Kodak).

### 2.6. Western Blotting and Coomassie Brilliant Blue (CBB) Staining

The inoculated *N. benthamiana* leaves were collected and homogenised in liquid nitrogen, followed by adding equal volumes of protein loading buffer, as described previously [29]. The samples were vortexed and boiled in distilled water for 10 min, then centrifuged at 12,000× *g* for 10 min. The supernatant was loaded to the 12.5% SDS-PAGE and separated by electrophoresis. The gel was stained by Coomassie brilliant blue (CBB) and decolourised with 10% acetic acid.

Western blot analyses were performed, as previous described [29]. The proteins on the gel were transferred to nitrocellulose membranes (Hybond-C, GE Healthcare). The membrane was blocked, incubated with a P1^SCSMV^-specific primary antibody, followed by incubation with a secondary antibody conjugated to protein A-alkaline phosphatase (Sigma-Aldrich, Shanghai, China). The antibodies used for analysis of endogenous proteins (RP-1, CAT, GST, L-ASO, LBC, and ARC5) were purchased from Agrisera (Olink^®^ Group, Uppsala, Sweden), and the commodity codes were AS10 687S, AS09 501, AS09 479, AS08 368, AS13 2709, and AS13 2676, respectively. The colour reaction was developed by adding the substrate solution (NBT and BCIP). The results were scanned by a scanner (CanonScanLiDE300, Canon, Japan).

### 2.7. Real-Time PCR Quantification of the Gene Expression

The TRIzol Reagent (Thermo Fisher Scientific, Shanghai, China) was used for total RNA isolation from plant leaves, and the cDNA was obtained from 1.0 μg RNA by reverse transcription reaction using M-MLV reverse transcriptase (Sigma-Aldrich, Shanghai, China). The quantification of gene expression analyses was performed on a CFX96 Touch Real-Time PCR Detection System (Bio-Rad, Hercules, USA) based on SsoFast EvaGreen Supermix (Bio-Rad, USA), as described previously [30]. The obtained cycle threshold (Ct) values were automatically calculated by the Bio-Rad CFX Manager 3.0 software. The *NbEF1a* gene was chosen as the internal control for the candidate gene expression analyses. The primers used in the study of gene expression are listed in Appendix A. The reactions were performed in triplicate, and the results were averaged.

### 2.8. Northern Blotting and DIG-Labelled PVX Specific DNA Probe Preparation

We designed the primers PVX/CP/F (5′-GCTGTCGCAACAAATGAGG-3′) and PVX/CP/R (5′-GTCGTTGGATTGTGCCCTG-3′) for amplification of the CPP^VX^-specific cDNA fragment. DIG-11-dUTP (Roche) was purchased and mixed with equal amounts of dATP, dCTP, and dGTP, and the final concentration of each was 10 mM. We used the prepared dNTPs for PCR and obtained the specific DIG-labelled DNA probe.

Northern blot was performed, as previously described [29], with minor modifications. Leaves infiltrated with *A. tumefaciens* were harvested, and total tissue RNA was isolated. The RNA was quantified with a NanoDrop ND-1000 (Thermos Fisher Scientific), which was denatured before loading to the 1.2% agarose/1.1% formaldehyde gel. We used 5 μg of total RNA that was separated by electrophoresis and transferred to Hybond-N^+^ membranes (GE Healthcare, USA). The membranes were fixed by UV crosslinking and stained with methylene blue. The prepared CP^PVX^-specific DIG-labelled DNA probe was added to the prehybridisation solution and hybridised overnight at 65 °C. After washing three times with SSC solution, the membrane was blocked with 1.2% evaporated milk, followed by incubation with the DIG-AP conjugated antibody (Merck). The colour reaction procedure was performed in the same way as previously described in the Western blot analyses.

### 2.9. Extraction of Total Proteins from Plant Tissue

Samples were homogenised in liquid nitrogen, after which the powder was transferred to a 5 mL centrifuge tube containing an equal volume of lysis buffer (1% TritonX-100, 10 mM dithiothreitol, and 1% protease inhibitor cocktail, 50 μM PR-619, 3 μM TSA, 50 mM NAM, and 2 mM EDTA). Thereafter, samples were sonicated three times on ice for 30 min using a high-intensity ultrasonic processor (Scientz), after which an equal volume of Tris-saturated phenol (pH = 8.0) was added and the samples were vortexed for 5 min. After centrifugation (4 °C, 10 min, 5000× *g*), the upper phenol phase was transferred to a new centrifuge tube. Total proteins were precipitated by ammonium sulfate-saturated methanol and incubated at −20 °C for at least 6 h. After centrifugation (4 °C, 10 min, 5000× *g*), the supernatant was discarded, and the precipitate was washed with ice-cold methanol once and ice-cold acetone three times. The precipitate was redissolved in 8 M urea. The concentration and the quantity of extracted protein was determined using a BCA kit (Thermos Fisher Scientific) and SDS-PAGE electrophoresis, respectively.

### 2.10. Digestion, LC-MS/MS, Database Searching, and Bioinformatics Analysis

The prepared plant total protein was reduced with 5 mM dithiothreitol for 30 min at 56 °C and alkylated with 15 mM iodoacetamide for 20 min at 56 °C in darkness. The protein samples were diluted with 100 mM TEAB to reach a urea concentration of less than 2 M. Trypsin was added to the protein samples according to the ratio of 1:100 (*m*/*m*) for digestion overnight.

The tryptic peptides were dissolved in 0.1% formic acid (solvent A), directly loaded onto a homemade reversed-phase analytical column (25 cm length, 75 μm i.d. 1.9 µm) for separation using a NanoElute UPLC system (Bruker, Salbrucken, Germany). The solvent B (0.1% formic acid in acetonitrile) gradient was composed of an increase from 4% to 6% over 2 min, 6% to 25% in 5 min, 25% to 32% in 5 min, 32% to 80% in 5 min, then climbing to 80% in 3 min before holding at 80% for the last 3 min, all at a constant flow rate of 280 nL/min.

The separated peptides were subjected to capillary ion source for ionisation, followed by timsTOF Pro (Bruker, Germany) analyses. The electrospray voltage applied was 1.4 kV. Peptide parent ions and their secondary fragments were detected and analysed with the TOF system. The m/z scan range was 100 to 1800 for a full scan. The data acquisition mode selected was Parallel Accumulation Serial Fragmentation (PASEF). Once the first mass spectrometry spectrum was obtained, the MS-MS spectrum was generated by 20 times PASEF data collection of parent ions with charge numbers ranging from 0 to 5, with 24 s dynamic exclusion to avoid repeated scanning.

The raw data acquired from the Bruker timsTOF Pro were subjected to Progenesis QI for proteomics v2.0 (NonLinear Dynamics). The Maxquant search engine (v.1.6.6.0, Max Planck Institute, Munich, Germany, https://www.maxquant.org/, accessed on 9 September 2022) was used to process the MS/MS data. The searching parameter settings were as follows: *N. benthamiana* reference genome, 76,474 proteins, https://solgenomics.net/, accessed on 9 September 2022.

The resulting MS/MS data were processed using the Maxquant search engine (v.1.5.2.8, cox and Matthias Mann, Max Planck Institute, Germany). Tandem mass spectra were searched against the *N. benthamiana* protein database (https://solgenomics.net/, accessed on 9 September 2022) and Uniport database, concatenated with the reverse decoy database. The FDR (false discovery rate) was adjusted to 1% to decrease the false positive rate, and the common pollution data were added to the database to remove the influence of the contaminations. Trypsin/P was specified as the cleavage enzyme, allowing up to two missing cleavages. The mass tolerance for precursor ions was set as 40 ppm in the first search and 40 ppm in the main search, and the mass tolerance for fragment ions was set as 0.04 Da. The FDR of the protein and peptide-spectrum identification matches was set as 1%.

### 2.11. Generation of P1^SCSMV^ Transgenic N. benthamiana Plants

*Agrobacterium* strain EHA105 containing the plasmid pGWB14-P1^SCSMV^ was prepared. Then, the leaf disc transformation method was applied to generate transgenic plants [31]. The cultivation, regeneration, and genomic DNA extraction of the explants, as well as PCR analysis, were performed as previously described [29].

### 2.12. Statistical Analysis

Student’s *t*-test and Tukey’s *post hoc* test was carried out to determine statistical significances in protein relative expression levels between samples. For quantification of proteins, the Progenesis LC-MS software was used for analysing the protein abundance values. The abundance of a protein was considered to be increased or decreased when the abundance was, respectively, higher (>1.50-fold change) or lower (<0.67-fold change) with or without P1^SCSMV^ expression by PVX.

The Image J software (v2.0, National Institutes of Health, Bethesda, USA, http://imagej.nih.gov/ij/, accessed on 9 September 2022) was used to quantify the intensity of the WB. The reaction colour in the PVDF membranes was determined according to the manufacturer’s instructions. Three membranes from independent experiments were measured. Each membrane was measured three times, and the results indicate the average intensity of the band colour that appeared.

## 3. Results

### 3.1. Heterologous and Cis-Expression of P1^SCSMV^ Enhances Viral Pathogenicity

The biological functions of P1^SCSMV^ were explored using a virus-derived overexpression vector based on PVX. The chimeric viruses PVX-GFP and PVX-P1^SCSMV^ were obtained by inserting the opening reading frames of GFP and P1^SCSMV^ into the viral overexpression vector. The agrobacterial inoculation medium, containing either PVX-GFP or PVX-P1^SCSMV^ and ssGFP, was injected into the leaves of transgenic *N. benthamiana*, plant line 16c. At five days post inoculation (dpi), the PVX-P1^SCSMV^/ssGFP-injected leaves showed a strong fluorescent signal compared to those injected with PVX-GFP/ssGFP (Figure 1A). This indicates the activity of P1^SCSMV^ as a VSR that can suppress systemic RNA silencing by the host plant.

Western blot analysis with specific antibodies showed a heterologous expression of P1^SCSMV^ in the upper leaves, thereby suggesting that the chimeric virus PVX-P1^SCSMV^ achieved a systemic infection of *N. benthamiana* (Figure 1B). At eight dpi, the upper leaves were collected for the isolation of total RNA, which was used in Northern blot analyses with specific probes complementary to the *CP* gene of PVX. We found that genomic RNA was more abundant in *cis*-heterologously expressed plants injected with PVX-P1^SCSMV^ than in those with PVX-GFP (Figure 1C). Stunted growth and necrosis of inoculated leaves was observed in plants infected with PVX-P1^SCSMV^ compared to those infected with PVX-GFP (Figure 1D). Considered together, the PVX-induced *cis*-heterologous expression of P1^SCSMV^ enhanced the pathogenicity of the recombinant virus and aggravated viral disease symptoms in *N. benthamiana*.

### 3.2. Identification, Quantification, and Analyses of DAPs after 4D Label-Free LC-MS/MS

Recently, VSRs have been linked to an increasing number of new functions beyond their basic role as a suppressor of host antiviral responses [29,32,33]. To determine whether the new role of P1^SCSMV^ contributed to an enhanced virus accumulation in infected plants, the proteomic changes resulting from the observed *cis*-heterologous expression of GFP and P1^SCSMV^ were analysed. This was performed using a label-free LC-MS/MS-based quantitative proteomics approach. At eight dpi, the upper leaves of plants in both treatment groups were collected, and total proteins were extracted. The general workflow of the 4D label-free LC-MS-/MS-based quantitative proteomics approach developed for this study is shown in Figure 1D.

A total of 272,951 spectra were obtained, and 117,636 spectra were matched to the *N. benthamiana* protein database (Appendix A). After bioinformatics analyses and searching, 6235 proteins were identified. Among them, 4125 quantifiable proteins were detected (Appendix A). In the quality validation process, with average mass error controlled to below 0.02 Da, 27,317 peptides were detected, suggesting the high mass accuracy of the MS data (Appendix A). Principal component analysis and Pearson’s correlation coefficient displayed sufficient reproducibility of our experiment (Appendix A). All these results demonstrate that the proteomic sequencing data were credible and reliable.

Two volcanic maps were plotted to understand the identified DAPs. The total number of DAPs was 2843 and 122 when the fold change (FC) was set to higher than 1.5 or 2, respectively, and the *p*-value was less than 0.05. In group I (FC ≥ 1.5, *p* < 0.05), 140 proteins were upregulated and 143 downregulated (Appendix A). In group II (FC ≥ 2, *p* < 0.05), 69 proteins were upregulated and 53 downregulated (Appendix A). In summary, the results indicate that the physiological status of *N. benthamiana* was severely affected by the conditioning cis-expression of P1^SCSMV^.

To clearly visualise the identified DAPs (Appendix A), we took the proteins from group II to draw a heat map (Figure 2). The most abundant proteins were a 5-epi-aristolochene synthase-like isoform X1 (NbEAS, 57.252 FC), a pathogenesis-related protein (NbPR-1, 18.169 FC), and an endochitinase A (NbECA, 15.126 FC). The least abundant proteins were a xyloglucan endotransglucosylase/hydrolase protein (NbXTH, 0.208 FC), a 101 kDa heat shock protein (NbHSP, 0.211 FC), and a Dynamin-like protein ARC5 (NbARC5, 0.217 FC) (Figure 2).

The proteins were ranked based on the fold change abundance values. The parameters were set as fold change (FC) ≥ 2 and *p*-value < 0.05. Among the 122 DAPs, 69 were upregulated and 53 were downregulated.

### 3.3. COG/KOG Annotation and KEGG Enrichment Analyses of DAPs

To better understand the functions of the DAPs, a Clusters of Orthologous Groups of proteins/Eukaryotic Orthologous groups (COG/KOG) annotation analysis was performed. The COG/KOG analysis classified 72.8% of all DAPs into four categories and 21 groups as follows (Figure 3): (1) metabolism (35.3%), (2) information storage and processing (9.2%), (3) cellular processes and signalling (15.9%), and (4) uncharacterised (12.4%). In category 1, the majority of the proteins (25.4%) had high abundance, whereas 9.9% of the proteins had low abundance. Category 2 contained 7.4% high-abundance proteins and 1.8% low-abundance proteins. In category 3, 9.5% of proteins were low-abundance, and 6.4% were high-abundance. Finally, in category 4, 7.1% were low abundance proteins, and 5.3% were high abundance proteins. Additionally, the COG/KOG annotation analysis showed that energy production and conversion (12 high, 1 low), lipid transport and metabolism (13 high, 4 low), and amino acid transport and metabolism (13 high, 5 low) were significantly upregulated, whereas transcription (4 low, 1 high), translation, ribosomal structure and biogenesis (12 low, 3 high), post-translational modification, protein turnover, and chaperones (14 low, 10 high) were downregulated. These data suggest that P1^SCSMV^ accelerates energy metabolism, lipid and amino acid transport, and metabolism. Furthermore, it suppresses RNA and protein biogenesis and protein degradation in plants.

Additionally, the KEGG pathway analysis identified 17 pathways in which DAPs were involved. The most enriched pathways were selected and are listed in Appendix A. Most DAPs were involved in the mitogen-activated protein kinase signalling pathway, fatty acid degradation, phenylpropanoid biosynthesis, α-linolenic acid metabolism, biosynthesis of unsaturated fatty acids, photosynthesis (antenna proteins), and transport and catabolism (peroxisomes). Only two β-D-glucan exohydrolases showed low abundance in the phenylpropanoid biosynthesis pathway. Overall, these results suggest that P1^SCSMV^ activated the defence mechanisms of the host (e.g., the mitogen-activated protein kinase signalling pathway, and transport and catabolism by peroxisomes). Additionally, it promoted the transformation of fatty acids to unsaturated fatty acids or α-linolenic acids. The five identified antenna proteins indicate that P1^SCSMV^ increased the assimilating activity of light energy during photosynthesis.

### 3.4. Gene Ontology Enrichment Analysis

To refine the affection of cis-heterologous expression of the P1^SCSMV^ protein by PVX in *N. benthamiana*, gene ontology (GO) enrichment analyses of the host DAPs were performed. The DAPs were categorised into three groups: (1) biological processes, (2) cellular components, and (3) molecular functions. The detailed analyses of group 1 indicated that the overall metabolism of the host, single-organism processes and the cellular processes were severely affected (Figure 4A). These results are in line with those obtained by KEGG pathway analyses (fatty acid degradation, phenylpropanoid biosynthesis, α-linolenic acid metabolism, and biosynthesis of unsaturated fatty acids were highly abundant). In group 2, differential regulation of cells, membranes, organelles, and macromolecular complex exchange were affected. This indicates that the physiological state, cellular architecture, and membrane stability of the plant were altered by the heterologous expression of P1^SCSMV^ (Figure 4B). Group 3 showed that catalytic activity and binding were the most affected molecular functions (Figure 4C), suggesting that normal cellular activity was compromised.

### 3.5. Validation of the Selected DAPs by qRT-PCR and Western Blotting

The samples from the transcriptome analysis were used to validate the DAPs at the transcriptional level. Twelve proteins were selected (Appendix A), and gene-specific primers were designed (Appendix A). The results of the qRT-PCR analysis of the 12 genes corresponding to the selected proteins showed that the expression of six proteins was upregulated, whereas the expression of the remaining proteins was downregulated. In particular, 5-epi-aristolochene synthase-like (NbEAS-like; 57.2 folds), endochinase A (NbECA; 15.1 folds), and pathogenesis-related protein 1-like (NbPR-1-like; 18.1 folds) were significantly upregulated (Figure 5A–C, Protein Lane). The expression levels of catalase (NbCAT; 5.5 folds), glutathione S-transferase U9-like (NbGST; 3.7 folds), and L-ascorbate oxidase-like isoform X2 (2.8 folds) were also upregulated (Figure 5D–F, Protein Lane). The results of the qRT-PCR analyses showed the transcription level of these upregulated proteins was in line with the proteomic data (Figure 5A–F, RNA Lane). The six downregulated genes were as follows: lycopene beta cyclase (NbLBC; 1.5 folds), cell division protein FtsZ homolog 1 (NbFtsZ; 2.5 folds), psbP domain-containing protein 5 (NbPsbP like; 3.8 folds), dynamin-like protein ARC5 isoform (NbARC5i; 4.6 folds), aspartic proteinase (NbASP; 4.4 folds), and xyloglucan endotransglucosylase (NbXTH; 4.8 folds) (Figure 5G–L, left panel). Additionally, the qRT-PCR results showed that the transcription level of the downregulated proteins decreased (Figure 5G–L, right panel). Taken together, these results confirm that the expression of the proteins and their corresponding RNA transcription occurred simultaneously.

For further validation of the protein expression level, we purchased six specific antibodies against pathogenesis-related protein 1-like (NbPR-1 like), catalase (NbCAT), glutathione S-transferase (NbGST), L-ascorbate oxidase (NbL-ASO), lycopene beta cyclase (NbLBC), and dynamin-like protein ARC5 isoform (NbARC5i). Western blot analyses were performed, and the results showed that the expression of NbPR-1 like, NbCAT, NbGST, and NbL-ASO were upregulated (Figure 6A–D). The expression of NbLBC and NbARC5i proteins were downregulated (Figure 6E–F). Protein quantitative analysis results strongly agreed with the proteomic data using the ImageJ software. In summary, Western blot results were in line with the qRT-PCR and proteomic data, and once again strongly evidenced that the qualitative proteomic data are convincing and reproducible.

### 3.6. Effects of Is-Heterogeneous Expression of P1^SCSMV^ on Intercellular Spread of PVX

To uncover the mechanism of the enhancing role of P1^SCSMV^ on PVX pathogenesis, we wanted to explain the mechanism from the perspective of the expression profile of the candidate interacting proteins of PVX-encoded proteins. Twenty-two candidate host proteins—including NbeEF1A (Eukaryotic translation elongation factor 1A) [34], NbeEF1Bβ1 (Plant translation elongation factor), NbeEF1Bβ2 [35], NbTIPs [36], NbFD1 (Ferredoxin 1) [37], NbREM1–NbREM4 [38], NbPlastocyanin [39], NbActins [40], NbPCIP1-1–NbPCIP1-3 [41], NbMPB2Cb-1, NbMPB2Cb-2 (TMV-MP30 binding protein 2C, MPB2C) [42], and NbCPIP1–NbCPIP4 (CP interacting proteins, CPIPs) [43]—were selected for further analyses of their expression and transcription levels in the presence and absence of P1^SCSMV^ in *N. benthamiana*.

The NbeEF1A binds to the p33 replicase and the 3′ end *cis*-acting element of tomato bushy stunt virus RNA, promoting viral genomic RNA replication by stabilising the replication complex [34]. The NbeEF1A and NbeEF1Bγ act synergistically for tombusvirus replication by stimulating viral negative-strand RNA syntheses [44]. The NbeEF1Bβ1 and NbeEF1A were found to directly interact with the triple gene block 1 (TGB1) of potato virus X, promoting viral infection of pepper and *N. benthamiana*. From the qRT-PCR results, we know that the relative expression levels of plant translation elongation factor-related factors (*NbeEF1A*, *NbeEF1Bβ1*, and *NbeEF1Bβ2*) were not altered (Figure 7A,B), which implies that these factors might not be affected in P1^SCSMV^ overexpression conditions.

The biological function of TIP has been linked with callose degradation of the cell wall, which can be recruited to the plasmodesmata by the TGB2^PVX^ protein to increase the cell size exclusion limit (SEL) and contribute to intercellular viral movements [34]. In our analyses, we found that the relative levels of *NbTIPs* were reduced under P1^SCSMV^ overexpression conditions (Figure 7C). Hence, the reduced *NbTIP* levels suggest that P1^SCSMV^ reduced the SEL-increasing roles of the TGB2 protein under PVX-P1^SCSMV^ infection conditions.

TGB1^PVX^ can directly interact with the NbFD1 that functions in the SEL of intercellular transmission, and the overexpression of NbFD1 interfered with the RNA silencing suppressor activity, which enhanced the resistance to PVX [37]. The qRT-PCR result showed that the NbFD1 expression level was decreased in the presence of the chimeric PVX-P1^SCSMV^ infection compared with the PVX-GFP infection (Figure 7D). Hence, overexpression of P1^SCSMV^ reduced the resistance of the host plant to PVX infection, which might increase the permeability of the plasmodesmata to enhance intercellular movement capability.

Remorin (NbREMs) is a resident protein located in the membrane rafts and plasmodesmata. Investigations showed that TGB1^PVX^ could physically interact with remorin, impairing virus intracellular movement through the plasmodesmata [38]. The qRT-PCR results showed that the relative expression levels of the four candidate *NbREMs* had different expression profiles within P1^SCSMV^ overexpression. Compared to GFP-overexpressed plants, *NbREM1, NbREM3*, and *NbREM4* were moderately downregulated, and *NbREM2* was slightly upregulated (Figure 7E). The downregulation of *NbREMs* by the overexpression of P1^SCSMV^ is beneficial for PVX intercellular movement and spread in *N. benthamiana*.

Plastocyanin is a chloroplast protein that functions in electron transportation in photosynthesis. It physically interacts with the CP^PVX^ using the N-terminal transit peptide, which affects the severity of symptoms in the plant [39]. The TGB1^PVX^ is capable of reorganising the actin/endomembrane and forming the virus replication factory, “X-body” [40]. Compared to a GFP overexpressed plant, the relative expression levels of *Nbplastocyanin* and *NbActin* were not changed in the presence of P1^SCSMV^ overexpression (Figure 7G). Hence, the PVX replication and intracellular movement were not affected by P1^SCSMV^ overexpression by means of the host factor plastocyanin and NbActins.

PVX CP-interacting protein 1 (NbPCIP1 or NbCPIP2) are plant-specific proteins that play an important role as susceptibility factors for viral infections or affect viral replication and movement [41,43]. The qRT-PCR results showed that the expression levels of *NbPCIP1-1*, *NbCPIP1*, and *NbPCIP1-2* were dramatically downregulated, while the expression levels of *NbCPIP2*, *NbCPIP3*, *NbCPIP4*, and *NbPCIP1-2* were not altered in chimeric PVX-P1^SCSMV^-infected plants compared with PVX-GFP-infected plants (Figure 7H,J). These results are contrary to the report that *NbPCIP1* and *NbCPIP* are viral replication susceptible factors. However, the published paper tested just one type of *NbPCIP1* and *NbCPIP*. In our study, there are several homologies of CP-interacting proteins. They could interact with the CP and viral RNA element simultaneously, thereby playing completely distinct functions in virus multiplication.

The PVX movement-associated proteins (TGB1s) can interact with the NbMPB2C protein (TMV-MP30 binding protein 2C) that functions in the controlling of the viral movement [42]. The qRT-PCR results showed that *NbMPB2Cb-1* and *NbMPB2Cb-2* were dramatically decreased in chimeric PVX-P1^SCSMV^-infected plants compared to PVX-GFP-infected plants (Figure 7I), which strongly suggests that the control of the viral movement was impaired by the downregulation of *NbMPB2C*, and chimeric PVX-P1^SCSMV^ movement was enhanced in *N. benthamiana*.

Taken together, *cis*-heterologous expression of the P1^SCSMV^ significantly reduced the relative levels of *NbTIPs*, *NbFD1*, *NbREM1*, *NbREM3*, *NbREM4*, *NbPCIP1*, *NbPCIP2*, *NbMPB2Cb-1*, and *NbMPB2Cb-2*, while increasing the expression levels of *NbREM2* in chimeric PVX-P1^SCSMV^-infected plants compared to PVX-GFP-infected plants (Figure 7). All these results suggest that P1^SCSMV^ expression can dramatically alter the physiological status of the host plant and that expression levels of host factors that direct interactions with viral proteins also respond to viral infections. The up- or downregulation of these factors contributes to the optimal replication and movement of the chimeric PVX-P1^SCSMV^, which demonstrates the enhancing role of PVX accumulation levels and pathogenesis by P1^SCSMV^ in *N. benthamiana*.

### 3.7. P1^SCSMV^ Attenuates Photosynthesis by Damaging the PSII System

Upon SCSMV infection, the leaves of sugarcane exhibit mosaic and chlorotic streaks, and the photosynthetic activities of chloroplasts are severely affected [45]. Hence, sugar accumulation and yield in sugarcane leaves are greatly decreased with SMD caused by SCSMV infection [8]. In the present study, the DAPs of PVX-P1^SCSMV^ vs. PVX-GFP included many photosynthetic pathway-related proteins (Figure 2), and the KOG functional classification results showed that the metabolism category was the most significantly affected (Figure 3). The PsbE, PsbP, PsaD, PetB, and PetC proteins accumulated in high abundance in PVX-P1^SCSMV^ compared to PVX-GFP (Appendix A). Light-harvesting chlorophyll protein complex (LHC) activity was also upregulated, and Lhca1, Lhca3, Lhcb2, Lhcb3, and Lhcb5 were more abundant in PVX-P1^SCSMV^ than in PVX-GFP (Appendix A). Partial proteins related to the photosynthesis pathway were significantly increased with P1^SCSMV^ overexpression, which may directly lead to chloroplast dysfunction. To further dissect the chloroplast dysfunction roles of P1^SCSMV^, we cultured P1^SCSMV^ stable-expressing transgenic *N. benthamiana* plants and used photosynthesis-related parameters to test the functions of chloroplasts. The results indicated that photosynthesis declined in SCSMV-infected sugarcane and in P1^SCSMV^-overexpressed transgenic *N. benthamiana* plants, and that damage mainly occurred in the PSII system (Figure 8).

## 4. Discussion

### 4.1. General Outcomes

Increasingly, applications of proteomics and transcriptomics are used to identify the global changes of proteins or genes that are affected by abiotic factors or viral infections [26,46,47,48,49]. For proteomics, 2-dimensional electrophoresis (2-DE) was widely used in previous studies [50]. Later, the iTRAQ and label-free peptide quantitation methods were used instead of 2-DE to provide high quantitative accuracy [51,52]. Recently, the high degree of reproducibility and quantitative accuracy in the method named the 4-dimensional label-free quantitation method (4D-LFQ) exhibited powerful functions and wide application prospects by adding the ion mobility dimension to traditional label-free peptide quantitation analyses [53].

Taking advantage of the 4D-LFQ approach, the global proteomic changes by *cis*-heterologous expression of the P1^SCSMV^ by PVX compared to PVX-GFP infection were analysed. We identified 6235 proteins in total, of which 4125 could be quantified (Appendix A). Among these identified proteins, there were 283 DAPs (FC ≥ 1.5, *p*-value < 0.05) caused by P1^SCSMV^ heterologous expression, which are involved in energy production and conversion, lipid and amino acid transport and metabolism, transcription, translation, and protein turnover, or which function as chaperones (Figure 2 and Figure 3). The credibility and repeatability of the proteomic data were validated by RT-qPCR and Western blot (Figure 5 and Figure 6). Agrobacterium-mediated overexpression of ssGFP and PVX-GFP/PVX-P1^SCSMV^ revealed that P1^SCSMV^ acts as a VSR that suppresses the local and systemic RNA silencing of *N. benthamiana* (Figure 1). The relative expression levels of genes for host factors that directly interact with PVX-encoded proteins were analysed, and the results indicated that P1^SCSMV^ is capable of regulating their expression to provide better cellular conditions for PVX multiplication (Figure 7). SCSMV-infected sugarcane and the overexpression of P1^SCSMV^ in *N. benthamiana* simultaneously showed decreased photosynthesis parameters that mainly caused damage to the PSII system (Figure 8). We proposed possible mechanisms for the pathogenicity-enhancing and photosynthesis-reducing roles of P1^SCSMV^ on *N. benthamiana* (Figure 9). For the first time, our investigation provided the remodling roles of a virus-encoded single protein (P1^SCSMV^) to host physiological status, especially for viruses belonging to *Poacevirus*.

### 4.2. P1^SCSMV^ Protects the Viral Genomic RNA from Degradation by Host RNA Silencing

In this study, we used a PVX-derived viral vector to overexpress the gene P1^SCSMV^ in *N. benthamiana* and analysed the subsequent proteomic changes using the 4D-LFQ approach. The viral RNA accumulation levels in infected systemic leaves were measured, and we found that PVX genomic RNA was more abundant in PVX-P1^SCSMV^-infected plants than in control PVX-GFP-infected plants (Figure 1C). P1^SCSMV^ suppressed ssGFP-induced local RNA silencing, and PVX-P1^SCSMV^ infiltrated the host cells, leading to a strong fluorescent signal in tested leaves, which showed strong green fluorescence, whereas no fluorescence was observed in PVX-GFP-infected plants (Figure 1B). These results demonstrate that P1^SCSMV^ is a strong VSR, which has also been demonstrated in a previous study [8].

P1 suppression of host RNA silencing is attributed to its nonspecific double-stranded RNA (dsRNA) binding activity, which is a key characteristic of various VSRs [11,54]. In virus–plant interactions, the intermediate dsRNA during viral replication often triggers RNA silencing by the host. This acts as the major antiviral innate immune defence mechanism of plants [55,56]. To successfully establish infection in plants, some viruses often directly encode VSRs to interact with the silencing pathways, to counteract RNA silencing by the host [56] or to bind the dsRNA to inhibit the dicing or translation of viral RNA [55,57].

The P1 protein of the genus *Poacevirus*, which belongs to the family Potyviridae, showed strong RNA silencing suppressing activity [58]. The P1 protein of Triticum mosaic virus (TriMV), which is in the same genus as the SCSMV, could enhance the PVX RNA accumulation in *N. benthamiana* and act as a viral disease-enhancer [58]. In our study, we revealed the same function of SCSMV-encoded P1. However, how P1^SCSMV^ enhances chimeric PVX-P1^SCSMV^ infection is still obscure. In the present study, P1^SCSMV^ enhanced the virus accumulation levels and pathogenesis in *N. benthamiana* (Figure 1C,D), which is consistent with a previous study. In the presence of VSR encoded by other viruses, the PVX genomic RNA accumulated higher than single PVX infections, and the pathogenicity of PVX was exacerbated in *Nicotiana* spp. [59,60,61,62]. The heterologous expression of P1^SCSMV^ increased the PVX genomic RNA accumulation and pathogenicity (Figure 1C,D) in the present study. Considered together, we conclude that P1^SCSMV^ acts as a VSR that can protect the viral genomic RNA from degradation by host RNA silencing.

### 4.3. P1^SCSMV^ Provides Better Cellular Conditions for PVX Accumulation

In virus replication, the host factors that directly interact with PVX-encoded proteins, including NbPCIPs and NbCPIPs, and their relative expression levels, were changed following PVX-P1^SCSMV^ infection compared to PVX-GFP. Specifically, the expression levels of *NbPCIP1-1*, *NbCPIP1*, and *NbPCIP1-2* were dramatically downregulated (Figure 7H,J). The relative mRNA level of *NbPCIP1* in PVX-infected *N. benthamiana* increased, and genetic methods confirmed that NbPCIP1 plays critical roles in viral replication [41]. The subcellular localisation of NbPCIP1 is ER-associated granulate structures, which suggests these are the replication sites of PVX [41]. In contrast, *NbPCIP1-1, NbCPIP1*, and *NbPCIP1-2* decreased in P1^SCSMV^-overexpressed plants by the chimeric PVX. However, the relative degree of downregulation was compared to the PVX-GFP, not mock-treated plants. Therefore, our results strongly implied that P1^SCSMV^ plays critical roles in PVX replication. For *NbCPIP1*, *NbCPIP2*, *NbCPIP3*, and *NbCPIP4*, their relative expression levels were differently regulated in P1^SCSMV^-overexpressed plants by the chimeric virus, and *NbCPIP1* was dramatically downregulated (Figure 7J). This result also seems to contradict published data that PVX infections induce expression of *NbCPIP1* in plants [63]. The CPIPs were supposed to act as important susceptibility factors of PVX, possibly by recruiting heat shock protein 70 chaperones for viral assembly or cellular movement in *Nicotiana tobacco* [63], whereas the NbCPIP2a and NbCPIP2b interact directly with the viral RNA stem-loop structures and CPs in plants for enhancing viral genomic RNA replication [43]. However, the relative downregulation of the *NbCPIP1* was compared with the chimeric PVX-GFP, not with mock plants. Perhaps, under the conditions of the PVX infection, *NbPCIP1*-*1*, *NbPCIP1-2*, and *NbCPIP1* were upregulated compared to the mock-treated plants. All these results indicate the need for further, deeper investigations.

In virus movements, the relative expression levels of genes corresponding to the host factors that directly bind with the viral proteins, including *NbTIPs*, *NbFD1*, *NbREM1*, *NbREM2*, *NbREM3*, *NbREM4*, *NbMPB2Cb-1*, and *NbMPB2Cb-2*, were differently regulated after PVX-P1^SCSMV^ infection compared with the PVX-GFP (Figure 7C–E,I). The *NbTIPs* and *NbFD1* were factors that directly enhanced viral cellular transportation in plant cells [36,37]. Our results showed that the expression levels of these two host factors were slightly downregulated in P1^SCSMV^-overexpressed plants by the chimeric PVX, which were compared with the chimeric PVX-GFP infection. However, these results were not beneficial for understanding the optimal movement of PVX in the plant. Exploration of the deeper mechanisms behind virus replication and movement are essential for future research.

NbREMs are supposed to be raft proteins located in the detergent-insoluble membranes and can interact with TGB1^PVX^ in plants. They function in virus macromolecular trafficking [38]. In our analyses, the relative expression levels of *NbREM1*, *NbREM3*, and *NbREM4* were downregulated (Figure 7E), while the *NbREM2* was upregulated following PVX-P1^SCSMV^ infection (Figure 7E). The results imply that NbREMs were directly involved in the intercellular movement of PVX in plants. However, whether the four NbREMs are functionally redundant or complementary in PVX movement requires further investigation. The NbMPB2Cb was thought to interact with the PVX-associated movement proteins, and thus reduced the PVX cellular movement ability in plants [42]. The two homologous proteins, *NbMPB2Cb-1* and *NbMPB2Cb-2*, were dramatically downregulated in PVX-P1^SCSMV^-infected plants compared to PVX-GFP infections (Figure 7I). These results imply that the overexpression of P1^SCSMV^ promoted the cellular movement of PVX in the plant, although the results were consistent with enhanced movement roles. However, the expression of these host factors was not confirmed by the protein levels, and further research is necessary to determine the detailed functions of NbMPB2Cb in viral movement. Collectively, all the up- or downregulation of host factors that directly bind to the PVX-encoded proteins by *cis*-overexpression via chimeric virus PVX-P1^SCSMV^ played essential roles for PVX replication and movement.

### 4.4. Overexpressing P1^SCSMV^ Decreases Photosynthesis by Damaging the PSII System

Following SCSMV infection, the leaves of sugarcane exhibit mosaic and chlorotic streaks, and the photosynthetic activities of chloroplasts are severely affected [45]. Hence, sugar accumulation and yield in sugarcane leaves are greatly decreased with SMD caused by SCSMV infection [8], which is supported by the observation that P1^SCSMV^ transgenic *N. benthamiana* displayed a yellower phenotype than nontransgenic plants (Unpublished Data). In the present study, the DAPs of PVX-P1^SCSMV^ vs. PVX-GFP included many photosynthetic pathway-related proteins (Figure 2), and the KOG functional classification results showed that the metabolism category was the most significantly affected (Figure 3). The PsbE, PsbP, PsaD, PetB, and PetC proteins accumulated in high abundance in PVX-P1^SCSMV^ compared to PVX-GFP (Appendix A). Light-harvesting chlorophyll protein complex (LHC) activity was also upregulated, and the components of Lhca1, Lhca3, Lhcb2, Lhcb3, and Lhcb5 were more abundant in PVX-P1^SCSMV^ than in PVX-GFP (Appendix A). Partial proteins related to the photosynthesis pathway were significantly increased after P1^SCSMV^ overexpression, which may directly lead to chloroplast dysfunction. This finding is supported by the photosynthesis-related parameters (Figure 8). The results indicate that overexpressing P1^SCSMV^ decreases photosynthesis by damaging the PSII system.

Chloroplasts are the biosynthesis sites of several major phytohormones, secondary messengers, and ROS, and these products play essential roles in plant immunity and growth [64]. The PSII reaction system is an inevitable process in an oxygen-rich environment and undergoes degradation by membrane-bound FtsH metalloprotease in *Arabidopsis thaliana* [65]. A mutated FtsH2 (FtsH2^G267D^) directly lost its substrate-unfolding activity and perturbed the PSII protein homeostasis, which resulted in the upregulation of the salicylic acid signalling transduction [65]. Chemical blockage of the PSII reaction directly affects the ethylene-induced hypocotyl response, which is caused by the ethylene signalling transduction [66].

Chloroplast dysfunction resulting from P1^SCSMV^ overexpression may similarly produce nonfunctional bioactive molecules and even disrupt the phytohormone balancing system. Hence, the innate immunity mediated by phytohormones may be suppressed following P1^SCSMV^ overexpression. However, whether P1^SCSMV^ interferes with host photosynthesis and affects the normal production or transportation of phytohormones in SCSMV-infected plants still needs further investigation. The mechanisms revealing the effects of P1^SCSMV^ on chloroplasts, especially the PSII system-related factors and phytohormone production machinery, remain unexplained.

In summary, all these results demonstrated that P1^SCSMV^ could change the physiological status of the host plant by *cis*-heterologous expression by the chimeric virus PVX-P1^SCSMV^. The P1^SCSMV^ has three negative impacts on host immunity, and it optimised viral movement and replication of PVX. We purposed a model that illustrated the possible mechanism of the enhancing roles of P1^SCSMV^ in PVX multiplication. First, the P1^SCSMV^ acts as a virus-encoded RNA silencing suppressor (VSR), capable of suppressing the antiviral silencing of the host plant (Figure 9 ❶). Furthermore, P1^SCSMV^ was able to interfere with the functioning of photosynthesis in the host plant by damaging the PSII system, which resulted in chloroplast dysfunction and disorder of the phytohormones they produce. This may lead to downregulated phytohormone-mediated basal resistance of host plants (Figure 9 ❷). Moreover, P1^SCSMV^ affects the relative expression profile of the host factors that direct interactions with the PVX-encoded proteins, which may improve the replication and movement of PVX in the plant. P1^SCSMV^ provided a relatively friendly microenvironment for PVX multiplication and spread (Figure 9 ❸). All three points contributed to the observed enhanced PVX accumulation and decreased plant immunity.

## Figures and Tables

**Figure 1 cells-11-02870-f001:**
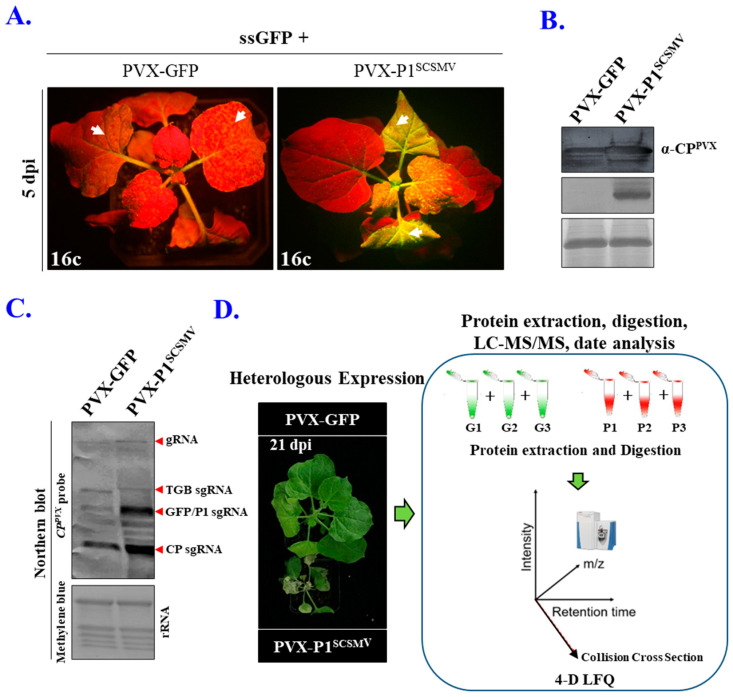
Disease-enhancing role of P1^SCSMV^ and the workflow of 4D-proteomics. (**A**) Validation of the RNA silencing suppression activity by *cis*-heterologously expressing GFP or sugarcane streak mosaic virus protein 1 (P1^SCSMV^) by potato virus X (PVX) on 16c transgenic *Nicotiana benthamiana* at 5 days post inoculation (dpi). The white arrows indicate the inoculated leaves. (**B**) Western blot was performed to detect the protein: upper panel: P1^SCSMV^-specific antibody; middle panel: CP^PVX^-specific antibody; bottom panel: large subunit of Rubisco was stained with Coomassie brilliant blue (CBB), which was used as a loading control. (**C**) Northern blotting was adapted to detect PVX accumulation in *cis*-heterologously expressed GFP and P1^SCSMV^ plants. Methylene blue-stained rRNA was used as a loading control. (**D**) Workflow for characterisation of the mechanisms of the disease-enhancing role of P1^SCSMV^ that is *cis*-heterologously expressed via PVX based on four-dimensional (4D) proteomics.

**Figure 2 cells-11-02870-f002:**
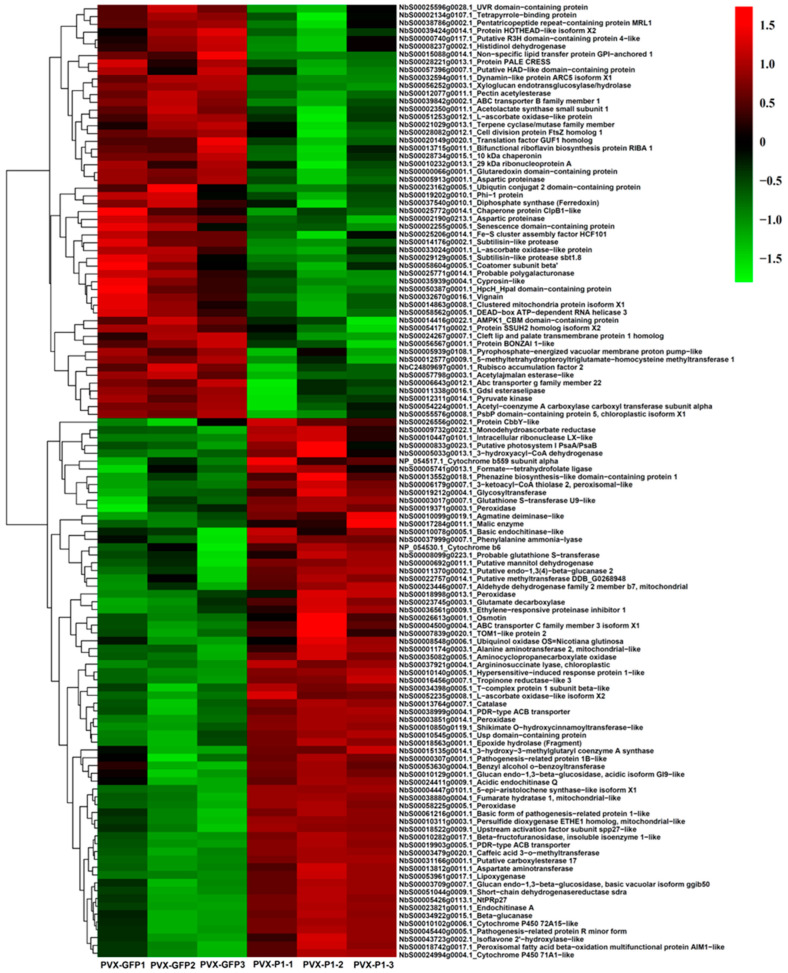
Heat map of the differentially accumulated proteins (DAPs) under heterogeneous expression of P1^SCSMV^ in *Nicotiana benthamiana*.

**Figure 3 cells-11-02870-f003:**
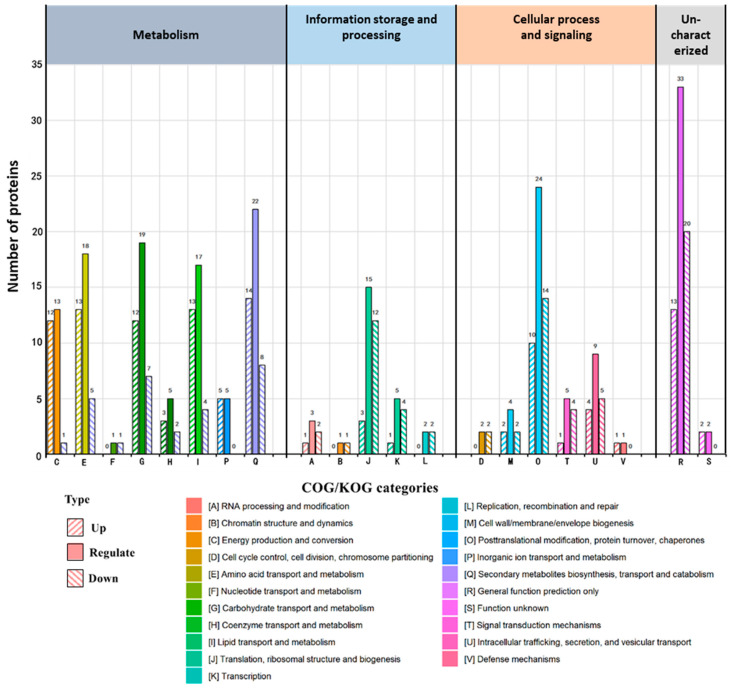
COG/KOG functional classification of differentially accumulated proteins grouped into 21 categories. The letters from A to V represent different categories according to the gene functions. Each category has three types of columns filled with solid colour and coloured slashes. The solid-colour-filled columns represent total regulated DAPs, and the coloured slash-filled columns represent up- and downregulated DAPs, respectively.

**Figure 4 cells-11-02870-f004:**
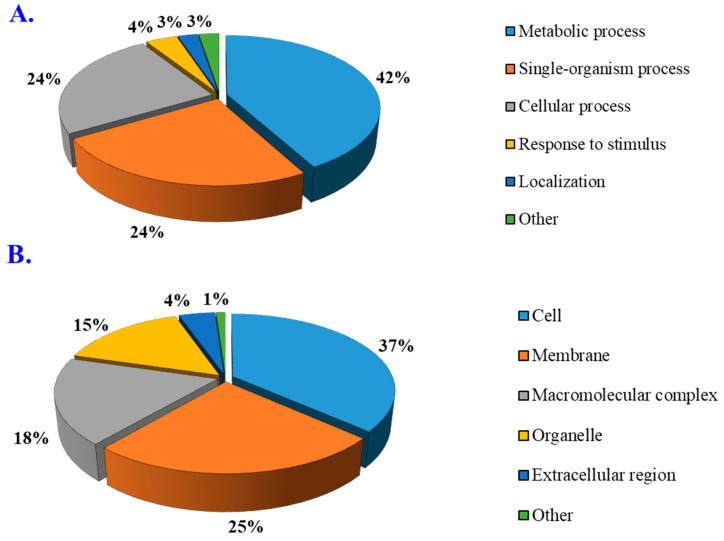
GO classification of differentially abundant proteins divided into three categories. (**A**) Biological processes. (**B**) Cellular components. (**C**) Molecular functions. Each category contains metabolic processes, single-organism processes, cellular processes, responses to stimuli, localisations, and others.

**Figure 5 cells-11-02870-f005:**
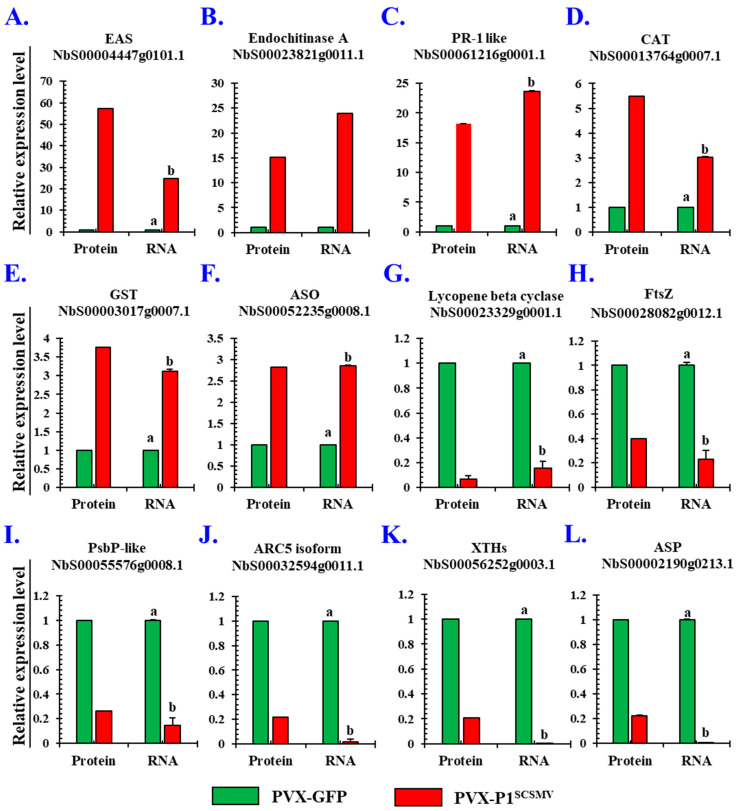
Validation of the relative expression levels of partial DAPs by qRT-PCR in *Nicotiana benthamiana.* Six upregulated and six downregulated DAPs were selected, and their corresponding mRNA expression levels were quantified by qRT-PCR analyses (left panel: protein expression level; right panel: RNA relative expression level). (**A**) 5-Epi-aristolochene synthase-like (EAS). (**B**) Endochitinase A (ECA). (**C**) Pathogenesis-related protein 1 like (PR-1 like). (**D**) Catalase (CAT). (**E**) Glutathione S-transferase U9-like (GST). (**F**) L-ascorbate oxidase-like isoform X2. (**G**) Lycopene beta cyclase, chloroplastic. (**H**) Cell division protein FtsZ homologue 1 (FtsZ). (**I**) PsbP domain-containing protein 5 (PsbP-like). (**J**) Dynamin-like protein ARC5 isoform (ARC5i). (**K**) Xyloglucan endotransglucosylase (XTHs). (**L**) Aspartic proteinase (ASP). Columns filled with green represent PVX-GFP-infected plants, whereas the red-filled columns refer to PVX-P1^SCSMV^-infected plants. The processed data were statistically analysed using SPSS software (version 22.0, IBM), and comparison analyses were performed by one-way analysis of variance (ANOVA). Different letters (a & b) mean significant difference between these two columns.

**Figure 6 cells-11-02870-f006:**
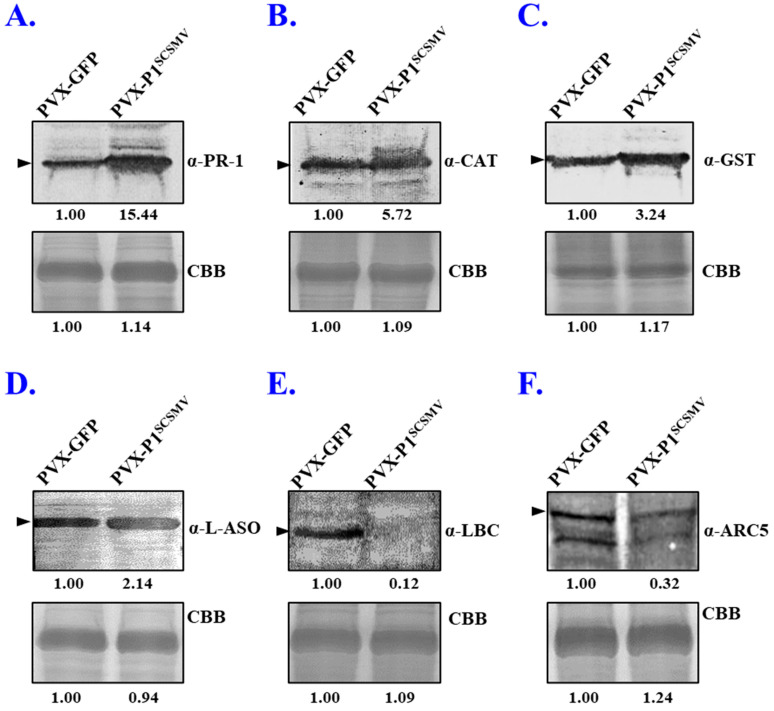
Validation of the relative expression levels of partial DAPs by Western blot in *Nicotiana benthamiana*. Six DAPs that showed different regulatory patterns (four upregulated and two downregulated) were selected for further validation of the reliability and reproducibility of the proteomics data. (**A**) Pathogenesis-related protein 1 like (PR-1 like). (**B**) Catalase (CAT). (**C**) Glutathione S-transferase U9-like (GST). (**D**) L-ascorbate oxidase-like isoform X2 (ASO). (**E**) Lycopene beta cyclase (LBC). (**F**) Dynamin-like protein ARC5 isoform (ARC5i). Each picture was replicated three times, and ImageJ software was used for protein expression level quantification. CCB staining was performed on the Rubisco large subunit, which was utilised as the loading control.

**Figure 7 cells-11-02870-f007:**
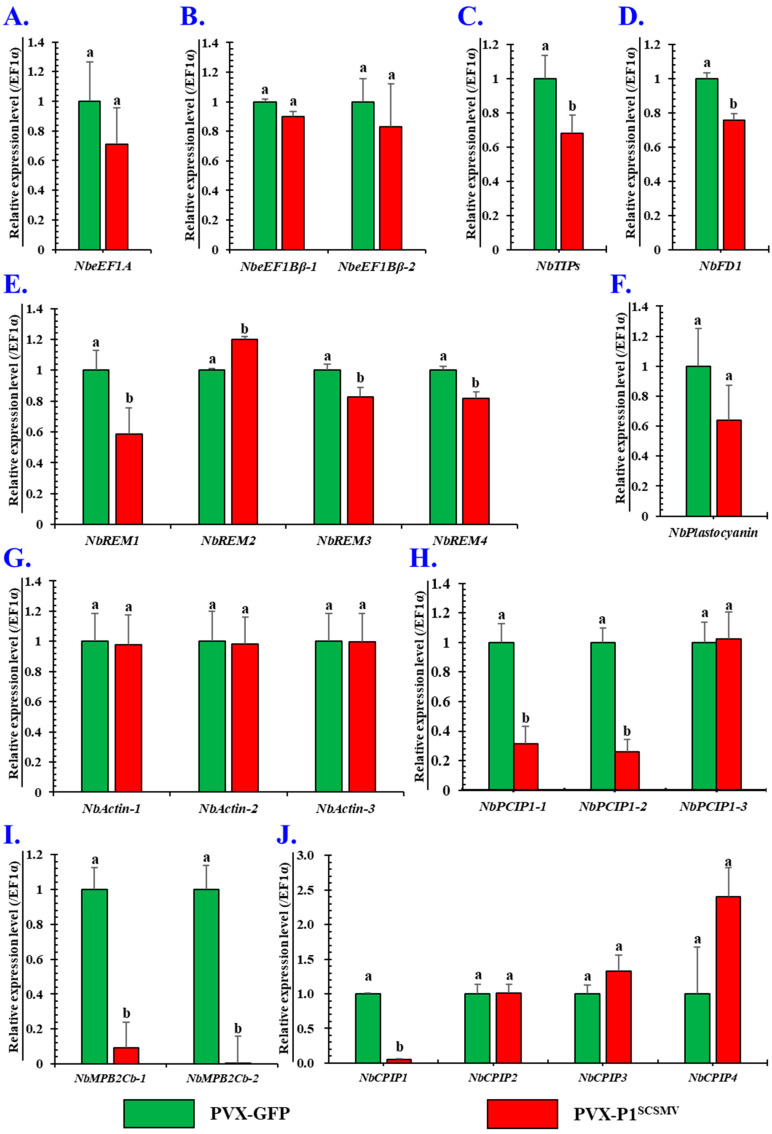
Relative expression levels of genes for ten candidate host factors that direct interactions with PVX-encoded proteins, as evaluated by qRT-PCR. Ten candidate genes, including *NbeEF1A* (**A**), *F1Bβs* (**B**), *NbTIPs* (**C**), *NbFD1* (**D**), *NbREMs* (**E**), *NbPlastocyanin* (**F**), *NbActins* (**G**), *NbPCIP1s* (**H**), *NbMPB2Cbs* (**I**), and *NbCPIPs* (**J**) interacted with the PVX-encoded proteins directly. The green columns refer to the PVX-GFP-infected plants, while red columns stand for the PVX-P1^SCSMV^-infected plants. The SPSS software was used for statistical analysis. Different letters (a & b) mean significant difference between these two columns.

**Figure 8 cells-11-02870-f008:**
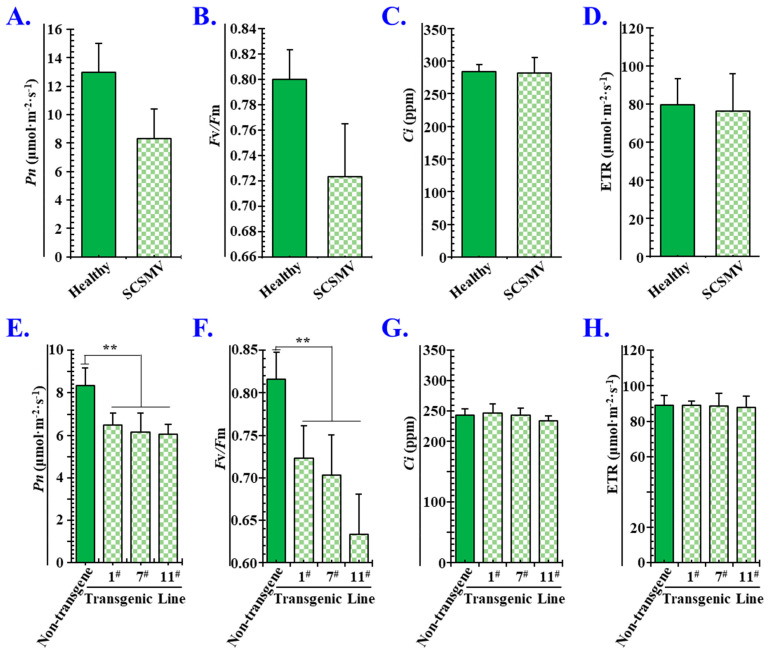
Effects of SCSMV infection and P1^SCSMV^ overexpression on photosynthesis. (**A**) Net photosynthesis rate (Pn). (**B**) Maximal quantum yield (Fv/Fm). (**C**) Intercellular CO_2_ mole fraction (Ci). (**D**) Electron transport rate (ETR) in SCSMV-infected sugarcane leaves. Each experiment was performed three times, and each treatment had at least three biological replicates. (**E**–**H**) Photosynthesis parameters Pn, Fv/Fm, Ci, and ETR in P1^SCSMV^ transgenic *N. benthamiana*, respectively. Error bars represent the standard error of the mean (SEM). Significant differences are indicated using a Student’s *t*-test: ** *p*-value < 0.01. # means different transgenic lines.

**Figure 9 cells-11-02870-f009:**
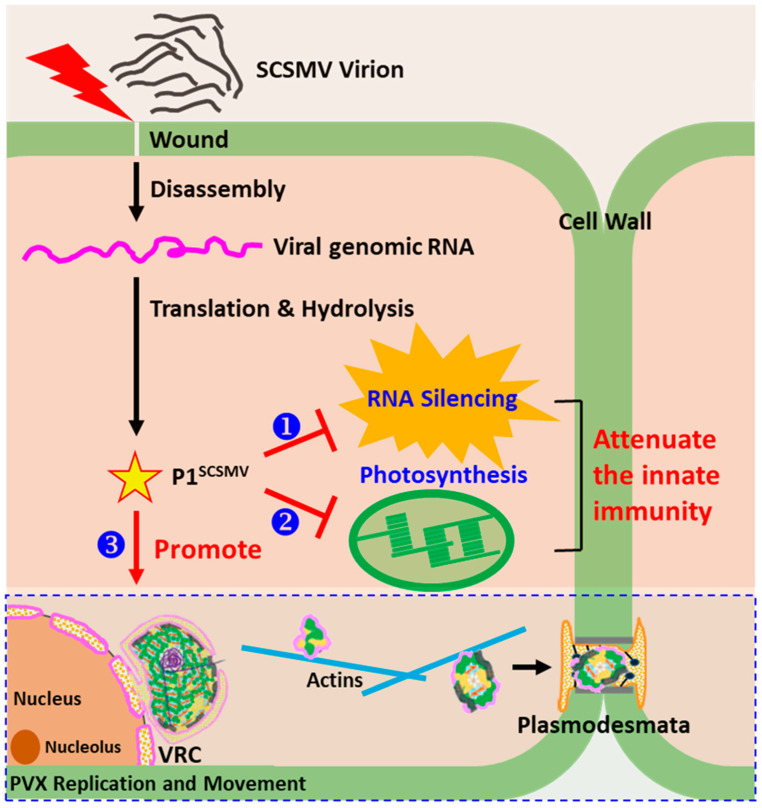
Schematic representation of the possible mechanisms of the disease-enhancing role of P1^SCSMV^ for PVX in plant cells. Under natural conditions, either the direct touching and rubbing of leaves or harvesting of sugarcane by cutting leads to SCSMV entry into the cell through a wound. After virion disassembly, naked viral genomic RNA could be utilised by the host for translation. The resulting large polyprotein could be directly hydrolysed by a self-encoded protease, and P1^SCSMV^ was consequently generated. In the present study, P1^SCSMV^ evenly distributed in the nucleus and cytoplasm could markedly attenuate the immunity of the plants through three aspects: ❶ P1^SCSMV^ acts as a VSR that directly suppresses host antiviral RNA silencing; ❷ P1^SCSMV^ inhibits the photosynthesic capability of the host, leading to dysfunctional chloroplasts and possibly reducing the phytohormone-based basal resistance of plants; and ❸ P1^SCSMV^ could up- or downregulate the host factors that directly interacted with the PVX-encoded protein to remodel the physiological status of the cell, which is conducive to PVX replication and movement. All these functions of P1^SCSMV^ clearly contributed to its PVX-mediated disease-enhancing role in plants. VRC, virus replication complex.

## Data Availability

The data presented in this study are available on request from the corresponding author.

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
