# Peer review of "Sugarcane Streak Mosaic Virus P1 Attenuates Plant Antiviral Immunity and Enhances Potato Virus X Infection in Nicotiana benthamiana"

_cells, 2022, doi:10.3390/cells11182870_

Round 1

Reviewer 1 Report

The manuscript entitled "Sugarcane streak mosaic virus P1 is involved in attenuating 2 plant antiviral immunity and enhances PVX infection on Nico- 3
tiana benthamiana
” describes the role
P1SCSMV in pathogenicity. The undertaken multidisciplinary approach not only established its role but rendered the mechanistic overview about the role of this protein. The manuscript is informative and data compilation is well justified with supported experiments. I would suggest authors to perform additionally to gain unbiased systematic overview of the involvement of DEPs in biological processes through MapMan analysis as a further validation of the GO and KEGG enrichment analyses.

Reviewer 2 Report

In this manuscript by Zhang et al, the authors take advantage of proteomics to explore new roles of P1 protein. They discovered many differentially accumulated proteins by P1 over-expression and validate them by qPCR and western blot. The findings are novel. However, this manuscript is very hard to understand due to typing and grammar errors. I strongly suggest the authors to polish the language with the help of professional companies. I also strongly suggest all the authors to double-check the typing errors. Additionally, discussion sounds like a repeated results description. Please rephrase and discuss extensively.

An excel table describing protein abundance is lacking.

Format is not uniform; some gene names are italic, and some are not. For example, Line 522-528: gene names are not italic. So is Figure 7 legend.

Line 512-516: “The qRT-PCR results showed that the expression levels of NbPCIP1-1, NbPCIP1, and NbPCIP1-2 were dramatically down-regulated, while the expression level of NbCPIP2, NbCPIP3, NbCPIP4, and NbPCIP1-2 was not altered in chimeric PVX-P1SCSMV infected plants compared with the PVX-GFP infected plant (Fig. 7H & 7J). This description is very confusing. It doesn’t match with the data in Fig. 7H and J. If labels in the figure are correct, I guess NbPCIP1-1 and NbPCIP1-2 are downregulated, while NbPCIP1-3 is not altered.

Line 553-557, the authors mentioned that “PsbE, PsbP, PsaD, PetB, PetC, Lhca1, Lhca3, Lhcb2, Lhcb3, and Lhcb5 were more abundance in PVX-P1SCSMV than in PVX-GFP”. Without the exact number, it is hard to evaluate the data in Supplemental Figure 2A and B.

About Figure 8, why don’t show the phenotype of P1 transgenic plants as the authors mentioned the yellow phenotype in Line 715. It is hard to believe that ETR was not altered when Fv/Fm was around or below 0.7. In the method section the authors cite Chen et al, 2017, but I did not find this article in references. This type of error would leave a bad impression to the reviewers. As the authors use Dual-PAM, I don’t know the ETR in figure 8 refers to ETR of PSI or ETR of PSII. Also, I couldn’t find any descriptions in Figure legends and methods. Please describe Figure 8 extensively in both Results and Method sections.

Line 715: “Unpublished date” should be “Unpublished data”. Line 124: it should be 267 micro mol·m-2·s-1, but not 267. mili mol·m-2·s-1. Frankly speaking, these typing errors should be corrected before submission. 

Reviewer 3 Report

Most plant viruses have profound and multiple-layer interactions with the host plant. To achieve robust replication and optimal infection, viruses often subverts host factors or directly suppresses the plant innate immunity. In the manuscript entitled "Sugarcane streak mosaic virus P1 is involved in attenuating plant antiviral immunity and enhances PVX infection on Nicotiana benthamiana” describes the role P1SCSMV in pathogenicity display. Overexpression of the P1SCSMV lead to many differentially accumulated proteins, and the results were validated by further qPCR and western blot. The present study describes the pathogenicity of P1SCSMV to plant cell via three aspects, especially the model proposed. The manuscript is informative and data compilation is well justified with supported experiments, and the findings are novel for sugarcane streak mosaic virus. However, there still exist some mistakes in writing, the language should be polished. Overall, I suggest to permit the publication of the present manuscript after some minor revisions.

Major revisions:

1.       Please to reorganize the reference, it seems that all the references are not your citied in the manuscript.

2.       There still exist some mistakes in writing, the language should be polished.

Round 2

Reviewer 2 Report

I didn't find any supplemental tables. There are only two supplemental Figures. Please double check whether you already submit the tables?

In general, the authors answered my questions except the missing tables.

In addition, the authors still need to check their format. Somewhere use Table S, somewhere use Supplementary Table. Also, different font were used in one sentence. Please check and revise all!

Author Response

Response to Reviewer #2's Comments: (2rd Round)

  1. I didn't find any supplemental tables. There are only two supplemental Figures. Please double check whether you already submit the tables?

Response: Thanks very much for your carefulness! Actually, we forgot to upload the Supplementary Tables, and we also could not find where to upload these files in these round of revision. I have already contacted with the editor, and emailed these files to her. Thanks again!

  1. In general, the authors answered my questions except the missing tables.

Response: Thanks very much for your carefulness and constructive suggestions!

  1. In addition, the authors still need to check their format. Somewhere use Table S, somewhere use Supplementary Table. Also, different font were used in one sentence. Please check and revise all!

Response: Thanks very much for your carefulness and constructive suggestions! we have checked the MS, and have revised these mistakes. Thanks again!